# A Telepresence Wheelchair with 360-Degree Vision Using WebRTC

**Van Kha Ly Ha \*, Rifai Chai and Hung T. Nguyen**

Faculty of Science, Engineering and Technology, Swinburne University of Technology, Hawthorn, VIC 3122, Australia; rchai@swin.edu.au (R.C.); hungnguyen@swin.edu.au (H.T.N.)

\* Correspondence: vha@swin.edu.au; Tel.: +61-404-042-6236

**Featured Application: The telepresence wheelchair system with 360-degree vision can use for people with disabilities in a variety of healthcare applications.**

**Abstract:** This paper presents an innovative approach to develop an advanced 360-degree vision telepresence wheelchair for healthcare applications. The study aims at improving a wide field of view surrounding the wheelchair to provide safe wheelchair navigation and efficient assistance for wheelchair users. A dual-fisheye camera is mounted in front of the wheelchair to capture images which can be then streamed over the Internet. A web real-time communication (WebRTC) protocol was implemented to provide efficient video and data streaming. An estimation model based on artificial neural networks was developed to evaluate the quality of experience (QoE) of video streaming. Experimental results confirmed that the proposed telepresence wheelchair system was able to stream a 360-degree video surrounding the wheelchair smoothly in real-time. The average streaming rate of the entire 360-degree video was 25.83 frames per second (fps), and the average peak signal to noise ratio (PSNR) was 29.06 dB. Simulation results of the proposed QoE estimation scheme provided a prediction accuracy of 94%. Furthermore, the results showed that the designed system could be controlled remotely via the wireless Internet to follow the desired path with high accuracy. The overall results demonstrate the effectiveness of our proposed approach for the 360-degree vision telepresence wheelchair for assistive technology applications.

**Keywords:** wheelchair; telepresence; assistive technology; 360-degree video; WebRTC

---

## 1. Introduction

Assistive technologies have played a vital role in supporting the elderly and people with disabilities. According to the global survey on assistive technologies from the World Health Organization, there will be more than two billion people needing at least one assistive product by 2030 [1]. This information is valuable in providing evidence-based resources that motivate researchers to meet the needs of people with disabilities. Over the last few decades, there have been significant achievements in assistive technology design and development. Apart from traditional wheelchairs, researchers have developed smart wheelchairs to improve the functionalities of the wheelchairs to meet the needs of the disabled people. Examples are an intelligent wheelchair for elderly people [2], smart wheelchairs which can avoid surrounding obstacles [3,4], and wheelchairs which can be controlled by using brain waves [5,6]. While the large proportion of previously existing studies about intelligent wheelchairs have primarily focused on developing a wheelchair controlled by a person sitting on it, there has been little research about the flexibility of controlling the mobility of wheelchairs while users are far from the wheelchairs.

In recent years, along with the explosive growth of telecommunications technologies, the field of telepresence has attracted very considerable interest in both academic and industrial fields.

Telepresence systems involve a combination of mobile robotics, computer, and telecommunication networks, which enable users to be virtually present and to interact in a remote location between the remote user and the local participants. A large proportion of telepresence robots were developed through mobile robot platforms [7,8]. Apart from telepresence robots for meetings, telepresence robots for education have also been explored [9]. Meanwhile, there has been limited work in telepresence healthcare, such as for remote visiting and assistance [10].

It is also important to highlight that the majority of the telepresence systems have been developed from robot platforms [8,11], while research on telepresence systems based on wheelchair platforms for mobility aids is still limited. Moreover, the benefit of telepresence as an assistive system for people with disabilities has not been much discussed in the literature. Although smart wheelchair systems currently exist, it is necessary to develop a new generation of smart wheelchairs, namely a telepresence wheelchair to fill this important gap.

Developing a telepresence wheelchair as assistive technology would be meaningful and would have great benefits for individuals who have mobility challenges. The telepresence wheelchair is a mobile wheelchair platform which is capable of providing two-way video communication. This system would have potential in a variety of healthcare applications and helpfulness for people with disabilities. One of the important applications is to provide support for independent mobility of the wheelchair user, especially in the scenarios when the wheelchair is located at a distance to the people with disabilities. For instance, people with disabilities are on the bed or somewhere that is far away from the wheelchair, and they might use the telepresence function to move the wheelchair towards their position without the help of caretakers. Thus, it is of practical interest to integrate telepresence function into the wheelchair to provide assistance for people with disabilities.

Moreover, the telepresence wheelchair can provide opportunities for social interaction. The people with disabilities, and individuals who have difficulty with mobility, may be restricted in terms of participating in communities in person due to lack of transportation. To aid those populations in reducing these barriers, the telepresence wheelchair can allow disabled people to control the wheelchair to move around in different places to meet people without actually sitting in the wheelchair. Such a system is also very helpful during the recovery and rehabilitation process. In these contexts, people with disabilities can avoid feelings of isolation and loneliness and, thus, their quality of life and health issues can be improved. Furthermore, remote monitoring and telehealth from the wheelchairs are common challenges for people with disabilities. Telepresence wheelchair systems are designed not only to provide assistance to the wheelchair users but also allow wheelchair users to interact with remote users such as healthcare professionals, nurses, as well as doctors for healthcare instruction. Moreover, telepresence wheelchairs can transmit the wheelchair user's biological signals such as information on heart rate, blood pressure, and body temperature for advanced monitoring in assistive healthcare applications. Individuals with disabilities would greatly benefit from these technologies. However, designing and developing such a telepresence wheelchair remains a technical challenge of real-time communication, remote control, and a wide field of view.

The primary objective of this research is to investigate the feasibility of a 360-degree field of view for a telepresence system and develop a telepresence wheelchair to provide assistive mobility for people with disabilities and older adults. A novel approach to emerging web real-time communication to improve system performance and develop an independent application is explored. The key idea of this unique telepresence wheelchair is to provide efficient and safe wheelchair navigation. Several techniques are involved in the process of completing the design in real-time applying real hardware. Finally, a complete telepresence system with a wide field of view and remote collaborate-control is designed. The aim is to not only provide a prototype for a telepresence wheelchair for healthcare assistants but also adopt the advancement of emerging information technology in the field of robotics, and biomedical engineering.

Our major contributions in this paper are summarized as follows. First, we developed and implemented the telepresence wheelchair with 360 degrees field of view. Then, we developed the real-time communication protocol of audio and video based on WebRTC with the acceptable performance of video quality. Finally, we developed the artificial neural network for QoE estimation of video

performance, which helps users can control the wheelchairs avoiding the region with the poor signal performance. The implemented wheelchair which was tested in various experiments demonstrated its applicability to telepresence functions.

The rest of the paper is organized as follows. Section 2 introduces the methodological approach. Section 3 is devoted to the experiments and results. Section 4 presents the discussions. Finally, Section 5 provides a conclusion to this paper and, then, sketches the directions for future research.

## 2. Methods

### 2.1. Telepresence Wheelchair with 360-Degree Vision Based on WebRTC

To enable two-way audio and video communication, we developed and implemented a telepresence wheelchair using a more flexible web real-time communication (WebRTC) framework. WebRTC is a standard that defines a set of communication protocols [12]. Some research about developing real-time applications has been started since the first version of WebRTC was released by Google [13]. Since then, much research has been developed to exploit the WebRTC application. For example, Chiang, et al. [14] presented a video conferencing system based on WebRTC. While web shopping using WebRTC was explored in [15]. WebRTC applications were also applied for education [16]. However, there has been no previous work which uses WebRTC for a telepresence wheelchair.

We have developed a telepresence wheelchair system based on an electric-powered wheelchair platform modified with many add-on external modules. The hardware of the system is illustrated in Figure 1. To achieve the desired features, the powered wheelchair is equipped with a Mac mini-computer, control components, telecommunication interfaces, touchscreen display. To enhance the full field of view, a camera is mounted in front of the wheelchair to capture the dual-fisheye images, and these two parts of the field of view are stitched into a single seamless panorama, and then streamed over the Internet. In particular, the Ricoh Theta S dual-fisheye camera is mounted in front of the wheelchair to provide coverage of the whole surrounding scene at the wheelchair location. As a result, a remote user can see a full view of the surroundings of the wheelchair to control the wheelchair more safely and efficiently.

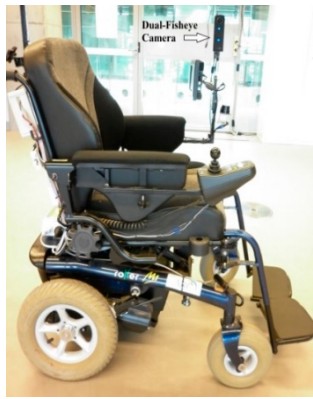

**Figure 1.** A telepresence wheelchair equipped with a dual-fisheye camera.

By developing an application programming interface and WebRTC-enabled browser, real-time video communications can be set up. Figure 2 shows 360-degree view telepresence using WebRTC architecture. A remote user can observe the surrounding environment of the wheelchair in real-time over wireless internet connections. From the obtained full field of view, a method for remote control approach to navigate the wheelchair from a remote location to interact with the remote environment must be developed.

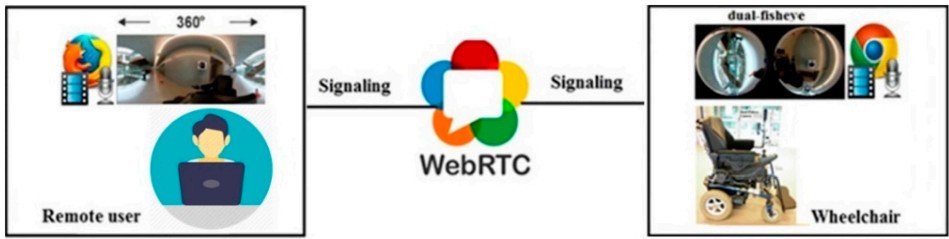

**Figure 2.** A 360-degree view telepresence using web real-time communication architecture.

For telepresence systems, one of the most important design objectives is to transfer voice, video, control signal, and data in real-time. The performance of data communications is carefully considered and thoroughly explored to understand the behavior of data transmission. The key factors affecting the performance of data transmission are bandwidth, throughput, round-trip-time, and frame rate. As a result, to evaluate the system performance, these important factors are analyzed.

The bandwidth (*W*) is closely related to the maximum communication channel capacity in bits per second. To determine the maximum rate of transmission of data which can be sent over a link, the bandwidth of a communications link is measured. If the bandwidth is sufficient, the packet delay is very low, and interactive media applications work quite well in terms of user satisfaction. The channel capacity is given by the Shannon–Hartley theorem as follows:

$$C = W \log_2(1 + \frac{S}{N}) \tag{1}$$

where *S/N* is a signal-to-noise ratio. For a system transmitting at maximum capacity, the bandwidth efficiency of the system can be written as

$$\frac{C}{W} = \log_2(1 + E_b \times \frac{C}{N_o \times W}) \tag{2}$$

where $E_b$ is the average received energy per bit, $N_o$ is the noise power density (Watts/Hz). Obviously, the larger the ratio $\frac{C}{W}$, the greater the bandwidth efficiency is.

On the other hand, the throughput is the number of bits transmitted per unit time. A packet contains L bits, and the amount of time devoted to that packet is the actual transmission time (*L/B*) plus the propagation delay (*d/V*).

$$Throughput = \frac{L}{\frac{d}{V} + \frac{L}{B}} \tag{3}$$

The TCP-friendly rate control (TFRC) bandwidth estimation model is used for bandwidth estimation algorithm, which relies on the p-value (the packet loss rate) [17].

$$T_{TFRC} = \frac{s}{RTT\sqrt{\frac{2p}{3}} + t_{RTO}\left(\sqrt[3]{\frac{3p}{8}}\right)p(1 + 32p^2)} \tag{4}$$

where *s* represents the average size of received packet size per second, *RTT* is round-trip-time and $t_{RTO}$ is the value of the retransmission timer. $T_{TFRC}$ does not represent the actual WebRTC real-time video output rate, but it represents an upper bound for it.

To deliver the video with improved quality, we adopted a Google congestion control (GCC) algorithm [13]. The sender-side uses the packet loss rate to estimate the sending rate (*As*) given by

$$As(i) = \begin{cases} \max\{S(i), As(i-1)(1 - 0.5p) & p > 0.1 \\ As(i-1) & 0.02 < p < 0.1 \\ 1.05(As(i-1) + 1 \text{ kbps}) & p < 0.02 \end{cases} \tag{5}$$

where *As(i)* is the sender available bandwidth estimate at time *i*, *S(i)* is the throughput, and *p* is the packet loss rate. The receiver-side estimates the receiving rate given by

$$Ar(i) = \begin{cases} \eta \, Ar(i-1) & \text{Increase} \\ Ar(i-1) & \text{Hold} \\ \alpha \, R(i) & \text{Decrease} \end{cases} \tag{6}$$

where $Ar(i-1)$ denotes the receiving rate estimate at time $(i-1)$, $\eta$ is the receiving rate increase constant. $R(i)$ is the current incoming rate, and $\alpha$ represents the incoming rate decrease factor.

### 2.2. Remote Control Telepresence Wheelchair via WebRTC

In the preceding sections, we have presented a unique telepresence wheelchair system with 360-vision video communication. To establish a real-time 360-degree vision wheelchair interaction, both the remote client and the wheelchair must retrieve a WebRTC-enabled web browser from a web application server. However, WebRTC only supports video communication and provides the capability for modifying various aspects of the application program interface (API) for developers to build independent applications. For instance, the users can make a modification related to speakers, microphones, video cameras, and input data. WebRTC does not include a mechanism for exchanging the control signal. Therefore, to control the wheelchair over WebRTC, a controller API must be developed to transfer the control signals from the remote user to the wheelchair.

In this work, the remotely controlling architecture via WebRTC for a telepresence wheelchair is depicted in Figure 3. As developed herein, a WebRTC media channel refers to a connection between two WebRTC clients for exchanging real-time media, while a WebRTC data channel relates to a connection for transferring data in one or more arbitrary formats. WebRTC data channel uses stream control transmission protocols (SCTP), which allows it to deliver and retransmit data. Moreover, WebRTC data channel API supports a flexible set of data types, such as strings and binary types. These types of data can be helpful when working with the transferring of a control signal from a remote user to the wheelchair. By using WebRTC data channel API, the control signal can be sent from a remote user to the wheelchair at the same time as the video communication. It is worth noting that a media channel and a data channel will be multiplexed and demultiplexed into a single peer connection between the WebRTC clients with the same user datagram protocol (UDP).

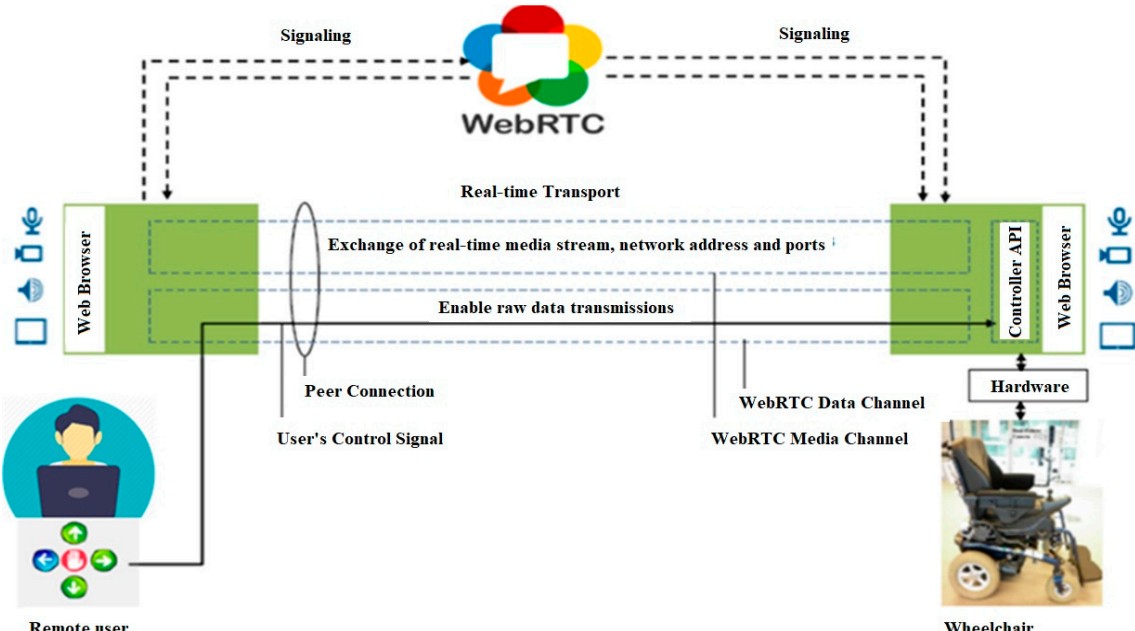

**Figure 3.** Remote control architecture of a telepresence wheelchair based on WebRTC.

To transmit control signals, we have developed an independent user-friendly interface with control buttons. The user interface permits a remote user to observe the surrounding environment of the wheelchair with audio and video communication and navigate the wheelchair by using the

control buttons at the same time and in real-time. The control signal originating from the remote user will be sent via the WebRTC data channel. At the same time, responsive to receiving the control signal from the remote user via the WebRTC data channel, the controller API at the wheelchair browser will modify functionality associated with the wheelchair computer and transfer the signals to hardware. In other words, the inputs of the controller API would be the data obtained from the remote user via the WebRTC data channel, and the outputs of the controller API would be the voltage values to control the wheel motors. By doing so, the framework will exchange information between hardware and software during the navigation process.

### 2.3. QoE Estimation of Wireless Video Streaming Using Neural Networks

In a wireless network-controlled telepresence system, the video transmission quality relies highly on the wireless network condition. Notably, the wireless signal fluctuations are affected by many factors and are difficult to forecast while the telepresence wheelchair is moving. The movement of the wheelchair into a weak wireless signal area can cause poor video streaming performance. Prediction of the quality of experience (QoE) is vital for a telepresence wheelchair in the short-term to react in time. However, the prediction of QoE while the wireless signal is not efficient to prevent loss of signal is a challenge. Thus, it is highly desirable to be able to predict the quality of service and, then, to properly control the wheelchair towards a wireless coverage area with strong signals.

The quality of experience evaluation of video can be either subjective or objective. However, subjective quality assessment methods for video are costly and time-consuming. Thus, objective evaluation is highly desirable to overcome these limitations by providing a mathematical calculation for the estimation of quality. Based on the availability of the original video signal, objective video quality assessment methods are categorized as full reference (intrusive) methods and free reference (non-intrusive) methods. Intrusive prediction models require access to the source, for example, peak signal to noise ratio (PSNR), structural similarity index (SSIM) whereas non-intrusive models do not and, hence, are more appropriate for real-time applications. From the literature, there are several other artificial intelligent techniques (machine learning-based) used to measure video quality, such as random neural networks, fuzzy systems, and artificial neural networks [18]. However, very little work has been done on predicting video QoE, considering the impacts of the fluctuations of wireless network conditions. Therefore, in this section, we develop a model based on artificial neural networks which are applied to predict the subjective quality metric of mean opinion score (MOS) for QoE estimation of video streaming using WebRTC over wireless networks. This section aims to present an estimation model based on neural networks for the objective, non-intrusive prediction of video quality over wireless streaming using WebRTC for video applications. To find an effective method to deal with uncertainty in the quality of service, we introduce an artificial neural network-based quality estimation model in streaming media applications that can estimate the quality of the video. Accurate QoE estimation of the system is necessary for the remote user to control the wheelchair towards reliable wireless signal coverages to maximize the QoE to meet the needs of real-time communication. The accuracy improvement of QoE prediction is a significant first step towards a more optimized feedback system that can efficiently control the system while moving the wheelchair into weak wireless signal coverage in the short-term.

In this section, we introduce the QoE estimation based on the Neural Networks (NNs), as shown in Figure 4. By exploiting the NNs, the measuring of the quality of the video can be done without accessing the original videos. After the NN is trained, the video quality evaluation can be performed in real-time.

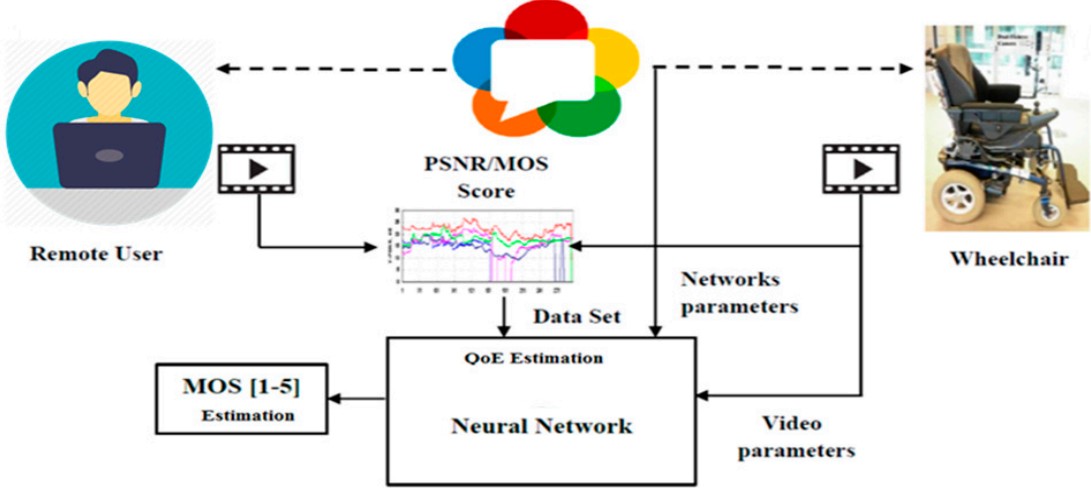

**Figure 4.** Quality of experience (QoE) estimation model based on the neural network.

This model uses a combination of objective parameters in the application and network layers, such as sender bit rate (SBR), jitter, packet loss rate (PLR), and round-trip time (RTT). The video quality was predicted in terms of the mean opinion score (MOS). With the aim of estimation of the video quality of experience, a neural network with a feed-forward three-layer topology is developed as shown in Figure 5.

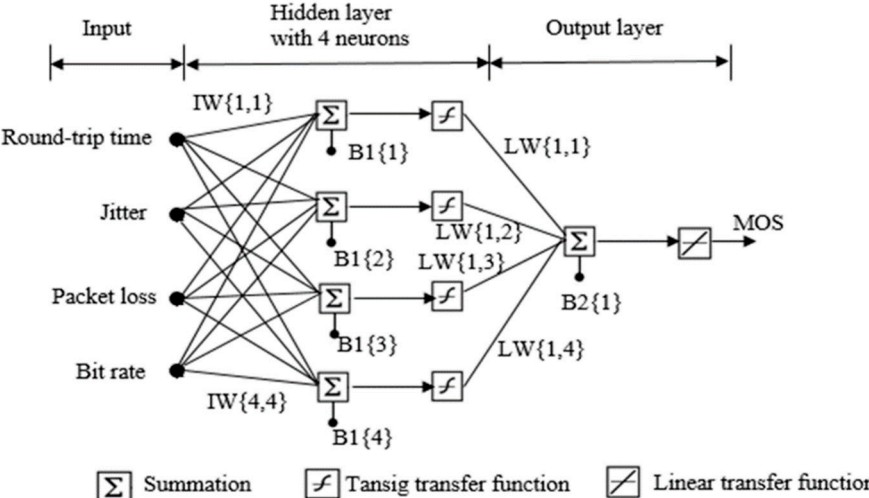

**Figure 5.** Block diagram of QoE estimation based on neural network.

There are two layers in the network, one hidden layer and one output layer. The input includes values of sender bit rate (SBR), jitter, packet loss rate (PLR), and round-trip time (RTT), and the output layer is the estimated QoE in terms of mean opinion score (MOS). MOS has commonly been used as the subjective metric during human subject experiments. To evaluate the video quality, The International Telecommunication Union (ITU) defined a five-level scale, called an ITU-5 point impairment scale, as shown in Table 1. To train the NN, the collection of MOS rankings in compliance with the ITU-T standard [18] during voting periods of human subject experiments was carried out.

**Table 1.** ITU five-point quality scale measurement of mean opinion score (MOS).

| MOS | Quality | Impairment |
|-----|---------|------------|
| 5 | Excellent | Imperceptible |
| 4 | Good | Perceptible but not annoying |
| 3 | Fair | Slightly annoying |

| | | |
|---|---|---|
| 2 | Poor | Annoying |
| 1 | Bad | Very Annoying |

As presented in [18], there is a close relationship between the MOS, the PSNR, and SSIM, as described in Table 2.

**Table 2.** Mapping the peak signal to noise ratio (PSNR), structural similarity index (SSIM) to the MOS.

| MOS | PSNR | SSIM | Quality |
|:---:|:---:|:---:|:---:|
| 5 | ≥37 | ≥0.99 | Excellent |
| 4 | 32–37 | 0.95–0.99 | Good |
| 3 | 25–32 | 0.88–0.95 | Fair |
| 2 | 20–25 | 0.55–0.88 | Poor |
| 1 | <20 | <0.55 | Bad |

After training the NN, the weights and bias parameters in the NN are determined and, then, the closed-form expressions for online prediction of user QoE can be obtained. The estimated QoE in terms of MOS is a function of online measurable network parameters (Jitter, SBR, PLR, and RTT) obtainable from wireless streams at monitoring points in the network. To develop the models, a series of well-designed objective and subjective test cases have been executed for the experiments. The ultimate goal of the objective and subjective testing is to come up with a set of user QoE training data from human subject experiments that can be fed to a neural network tool.

## 3. Results

We have discussed the techniques and all system design aspects related to our telepresence wheelchair system. It is possible to establish a complete system with such an approach. In this section, we present the experiments conducted in real-world environments and discuss the results to evaluate the system performance.

### 3.1. Experiment 1: To evaluate Real-Time Response of Telepresence Wheelchair System

The main aim of this experiment was to analyze the performance of WebRTC in various network conditions. In this experiment, we conducted the experiments consisting of ten trials. We evaluated the performance of the proposed method by implementing WebRTC to stream videos from the wheelchair to a remote location over the Internet. For each trial, the data transmission was observed during a period from 0 to 500 seconds. The remote user and wheelchair computer were configured with the Intel Core i7 CPU and 8 GB RAM to have sufficient computing power. Window operation system 7 (64 bit), Internet browser (Firefox) was installed. The bandwidth management was taken into account to ensure the consistency of the system performance. To examine the WebRTC performance, we evaluated the objective quality measures. We collected data from the peer connections between the sender and receiver for the cases when both the uplink and downlink bandwidths are unlimited and limited to 1500, 750, 250 kbps. The measurement data were recorded and collected carefully. A diagnostic packet and event recording related to the bandwidth estimation, packets sent, round-trip time and frame rates were measured and logged for analysis and evaluation.

**Results:** Figure 6 shows the distributions of average bit rates for the test cases with different bandwidths. The results show that when the bandwidth is limited, the bit rates are slightly lower than the controlled bandwidth for each video stream. As can be seen, it can reach the available bandwidth and maintain a constant bit rate. As expected, the system is designed to be able to use all the available bandwidth of Internet connections for video streaming and sending control signals. It seems that the remaining bandwidth, where test has a lower throughput, might be caused by application performance, for example, the web browser might be too busy to handle data transmission over data channel, other activities of the performance of the computer or the network condition has changed during the connection established between the sender and receiver. In contrast, it is observed that the highest

bit rate is 2500 kbps for the case of unlimited bandwidth. This is either limited by bandwidth estimation of the congestion control, or that the session does not need to send at a higher bit rate. In addition to measuring the bit rate, we also observed connections where the signal loss was significant. By continuously lowering the available bandwidth in additional experiments, we observed that a minimum of 250 kbps is necessary to establish video calls between participants. However, at least 250 kbps of available bandwidth is necessary to obtain a somewhat acceptable frame rate at the lowest possible video resolution (320 × 240 pixels). It takes a longer time to reach the maximum streaming rate, especially when the available speed is at the 250 kbps.

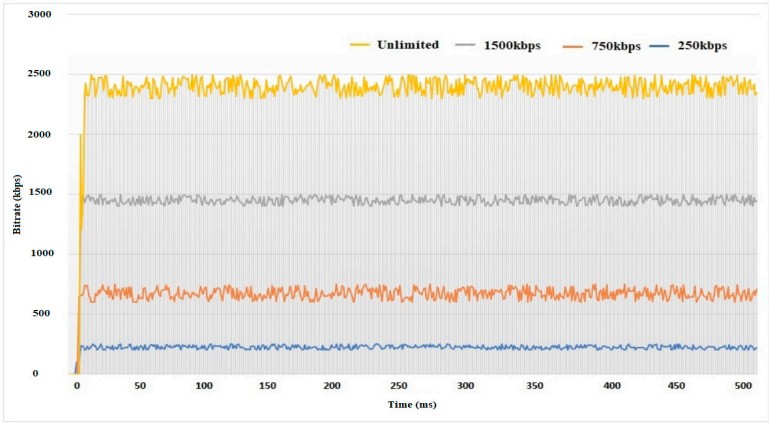

**Figure 6.** Bit rate with limited bandwidth and unlimited bandwidth.

Apart from being limited by a constant bandwidth, a more common scenario for real-time video streaming applications over wireless networks is that the network speed varies during the connection due to the nature of the fluctuation of the wireless signals and wireless channel conditions. In this experiment, we monitored the performance of how fast WebRTC adapts to new conditions in the real-world network condition. We observed this behavior by setting the maximum network available bandwidth up to 1 Mbps. It is notable that WebRTC's congestion algorithm does not respond directly to different latencies but changes its data rate based on latency variation. The results in Figure 7 show that the packets sending rate adapted with the available bandwidth and reached the maximum speed rate.

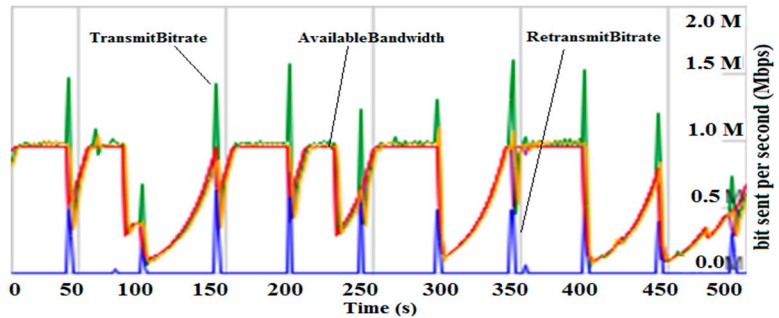

**Figure 7.** Available bandwidth and bit rate of the system.

The video transmission performance results of 10 trials are illustrated in Figures 8–10. As can be seen from Figure 8, the streaming rate of the video with the resolution of 1280 × 720 pixels varied and the average streaming rate was 25.83 fps. The results in Figure 9 indicate that the average round-trip-time (RTT) fluctuated from 3 to 20 ms. The results in Figure 10 show that the average peak signal-to-noise ratio is measured in the range of 28 to 36.5 dB.

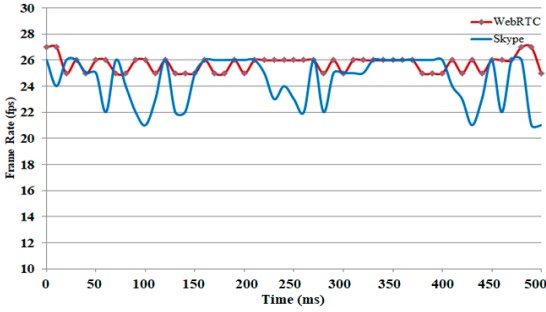

**Figure 8.** Frame rate sent (fps) of WeRTC versus Skype.

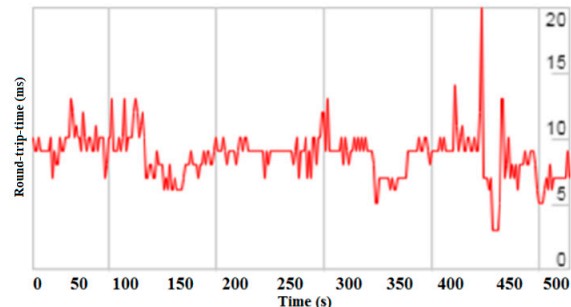

**Figure 9.** Round-trip time (RTT) of telepresence based on WebRTC.

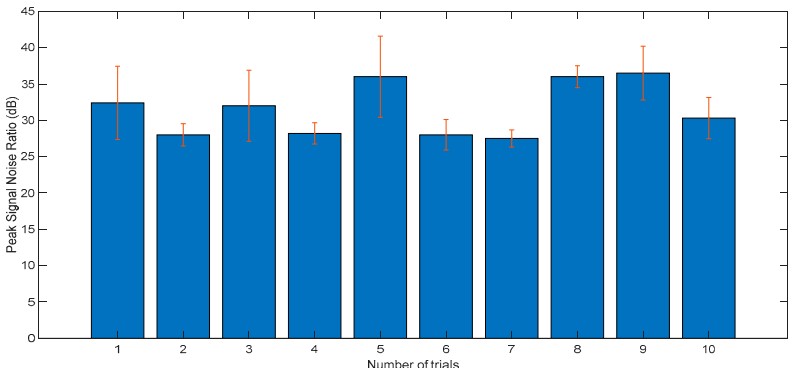

**Figure 10.** The peak signal-noise ratio of 10 experiments.

In comparison with [19], it is worth noting that the standard video frame rate is 25 fps (Europe) or 30 fps (North America). Therefore, the video streaming rate of the proposed system has reached the standard of real-time video communication. Regarding the round-trip time, it is worth noting that the RTT of less than 400 ms is considered as real-time communications [20]. The RTTs of WebRTC between two browsers in the experiments are under the RTT threshold value. The RTTs of the experiments are also significantly lower than the results of the previous telepresence robot study, which revealed that a 40 ms delay was considered good performance [21].

The results of 10 trials in Figure 10 indicate that the average peak signal to noise ratio (PSNR) was 31.49 dB. It is notable that the previous study of video streaming evaluation [22] showed that the quality of the image is satisfactory for 25 dB ≤ PSNR ≤ 32 dB and good for PSNR ≥ 32 dB. In comparison with [22], it can be concluded that the quality of the streaming video is satisfactory and the proposed approach of a 360-degree vision for the telepresence wheelchair is feasible.

Table 3 provides the summary of results of the processing time of the encoded video, the delay and the throughput for each resolution in all trials for different video resolutions. It can be seen from Table 3 that the average encoding time is increased when processing higher resolution videos. Delay during transmission may occur, which, in turn, will result in increasing buffer size for temporary storage, causing longer latency. However, a higher resolution results in more data to be transmitted but does not affect the throughput of the streaming due to congestion control. Overall, the experimental

results show that the streaming is smooth and seamless. The results demonstrate the effectiveness of our approach for the telepresence wheelchair with a wide field of view.

**Table 3.** Average video streaming characteristics for different video resolutions.

| Resolution (Pixels) | Average Encoding Time (ms) | Average Delay (ms) | Average Data Throughput (kbps) | Packet Loss (%) |
|---|---|---|---|---|
| 320 × 240 | 4.00 ± 0.01 | 8.06 ± 0.03 | 1641.635 | 0.10 |
| 640 × 480 | 7.00 ± 0.05 | 12.08 ± 0.16 | 1617.374 | 0.36 |
| 1024 × 576 | 11.00 ± 0.02 | 18.01 ± 0.23 | 1582.916 | 1.07 |
| 1280 × 720 | 11.46 ± 0.06 | 18.73 ± 0.18 | 1492.122 | 1.11 |

*3.2. Experiment 2: QoE Estimation of Wireless Video Streaming Using Neural Networks*

In this experiment, we developed a feed-forward neural network which consisted of multiple layers of neurons followed by an output layer of linear neurons. The model consists of an input layer, hidden layers, and an output layer. The output layer is of the Purelin type which covers the entire MOS range. The modeling scheme chosen is a feed-forward with a two-layer neural network. This gives the model the ability to approximate both linear and non-linear data functions. One hidden layer was used to perform the computation of weights and biases with considerations of the trade-off between performance and accuracy. It is known that the greater the number of hidden layers, the greater is the time taken by the model to compute the weights and bias. However, the weights and bias model the function very accurately.

The tansig function (covers range −1 to 1) is used to facilitate accurate modeling of non-linear aspects of the modeled data. The linear function is used at the output layer such that the output can take the entire MOS values.

In the training phase, it is important to define the number of epochs which is the number of sweeps through all the records in the training set. To have model accuracy, multiple sweeps are required. The number of epochs affects both the performance and accuracy of the model. With a large number of epochs, the model gets overtrained with given input set and does not correspond well to small fluctuations of input data. With a small number of epochs, model accuracy is reduced.

The type of training function used is the Trainlm (Levenberg–Marquardt), which is suitable in terms of accuracy for the size of the network in our problem. It cuts off the model training (number of epoch) before the model becomes overtrained with a given data set and learns much faster than the normal train function training. The neural network toolbox available in MATLAB is used for developing our QoE model estimation. The proposed neural network (NN) is presented in Figure 11. The parameters to be modeled are network jitter, round-trip-time, packet loss, and bit rate. The resulting output is the mean opinion score ranking with range (1–5).

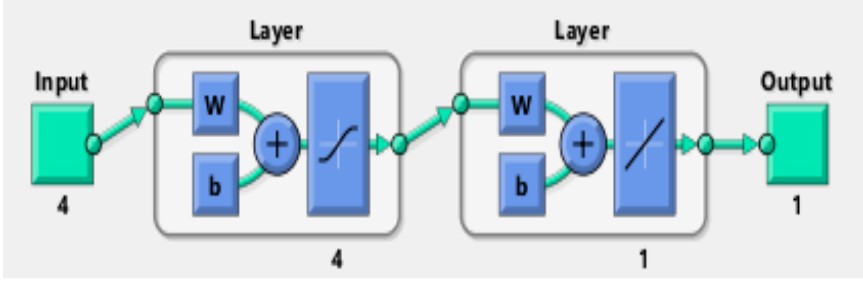

**Figure 11.** The proposed neural network for QoE estimation with weights w and bias b.

We collected the set of 300 cases in which MOS rankings are in compliance with the ITU-T standard. The NN was trained by using 100 cases while the other 100 cases were used for the NN validation, and the remaining 100 cases were used to test the NN.

**Results:** Figure 12 indicates the performance progress for training, validation and test over epochs. It is worth noting that the number of epochs is the number of sweeps through all the data records in the training set. The number of epochs is an important factor which determines the performance and accuracy of the NN model. It can be observed from Figure 10 that the best number of epochs is 12 where the validation performance reached a minimum.

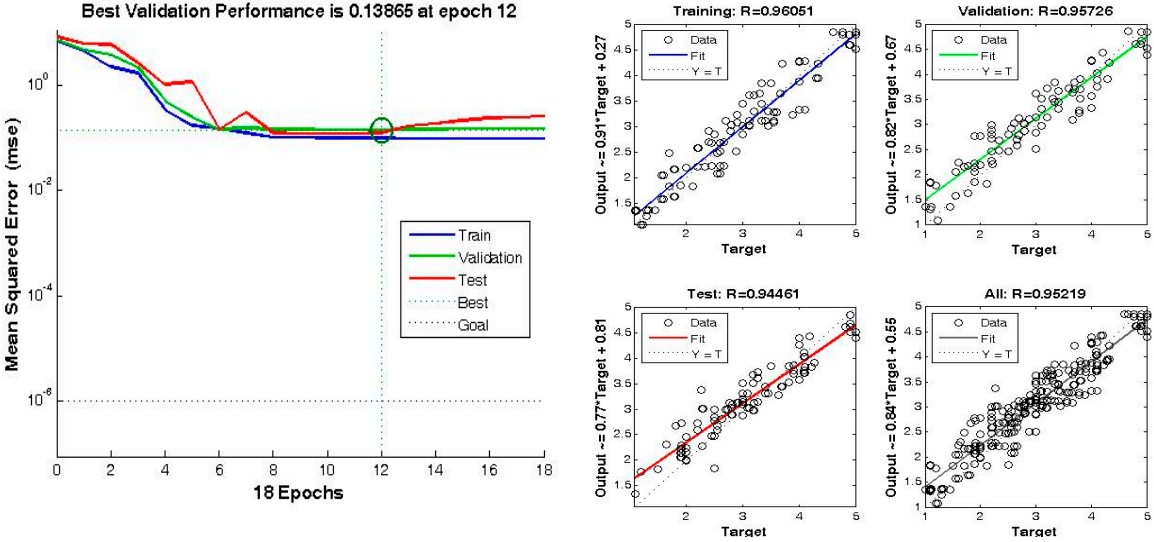

**Figure 12.** The performance progress over epochs and regression plots for the training, validation, testing, and all data.

To validate the neural network, we provide a regression plot in Figure 12, which reveals the relationship between the MOS output of the NN and the MOS target. As can be seen in Figure 12, the training data shows a good fit while validation and test results indicate R values that greater than 0.9. We used the trained NN to estimate the QoE for 100 test cases of video streaming. The estimated MOS is presented in Figure 12 in comparison with the measured MOS. The correlation between the estimated MOS and measured MOS is about 94%. The simulation results demonstrated that the proposed scheme provides good predictive accuracy (~94%) between the measured and estimated values. The results achieved reasonable prediction values from the neural network models. This work can potentially help in predicting the quality of service (QoS) in order to react in time for video streaming of a telepresence wheelchair over wireless networks.

The measured MOS in Figure 13 demonstrated that the video transmission quality in our telepresence wheelchair is satisfactory. The results show that good estimation accuracy was obtained from the proposed prediction model. This study should help in the development of an estimation of the video quality of a telepresence wheelchair. Moreover, by observing the estimated MOS score, the remote user can navigate the wheelchair into the area, which can guarantee a continuous connection and provide good performance.

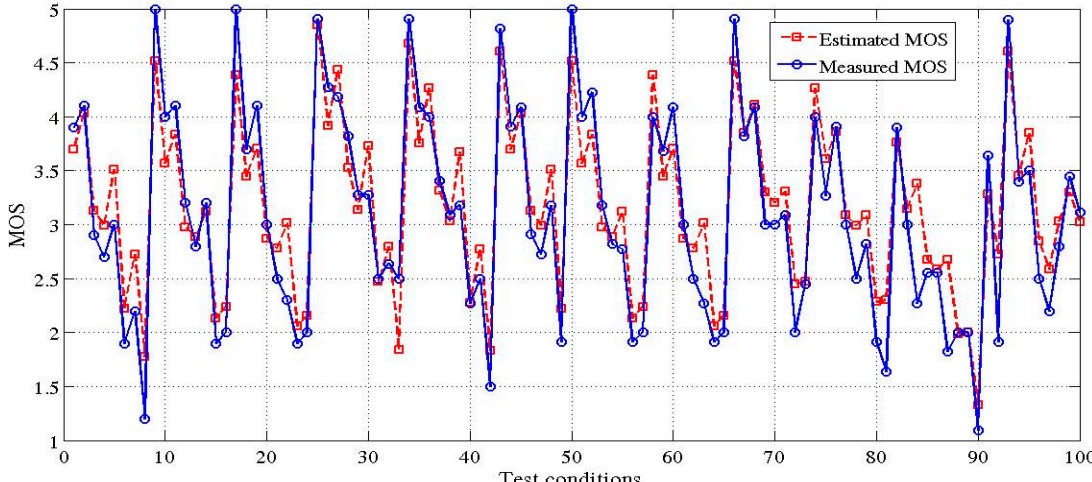

**Figure 13.** Estimated MOS based on Artificial Neural Network (ANN) versus measured MOS.

### 3.3. Experiment 3: Evaluation of Remote Control Telepresence Wheelchair

The purpose of the experiment was to assess the improvement of performance and to further demonstrate the effectiveness of the remote controlling function of our design. The tests were carried out by remote controlling the wheelchair in three locations in both indoor and outdoor environments. Ten experiments were conducted in each location. The average time spent on each test was two hours. The remote users were in a separate room where they could not see the paths and the targets, and they could only see them while driving the wheelchair through the telepresence function with 360-degree vision.

Firstly, the experiments were conducted in an indoor environment at the Centre for Health Technologies, where some free spaces were available. The tests were performed on the balcony where there is a wireless signal, as shown in Figure 14. This environment was considered to be completely unknown to the remote user in each operation time. The remote user was far away from the wheelchair and was required to control the wheelchair to reach specified targets.

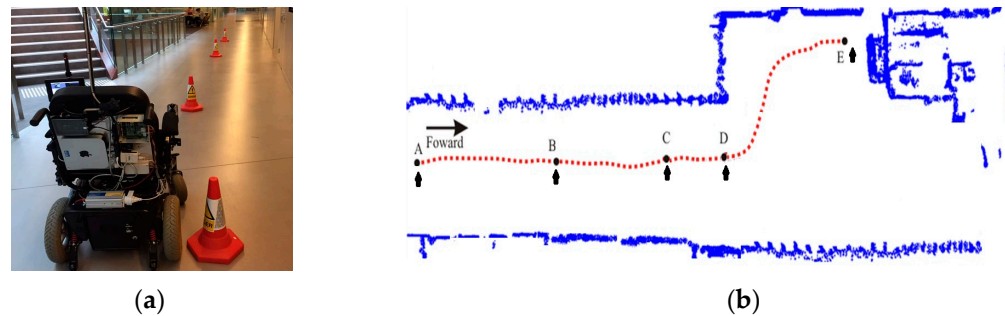

|     |     |
| --- | --- |
| (**a**) | (**b**) |

**Figure 14.** System performance test (**a**) and trajectory of the wheelchair when the remote user controlled the wheelchair with straight-ahead directions (**b**).

Figure 14 shows the environment and a trajectory produced by the wheelchair in the experiments. Starting from point A, the remote user controlled the wheelchair to move towards points B, C, D, and E, respectively, via the control of the remote user via a web browser. The wheelchair created a trajectory marked by the red dotted line. This trajectory is the result of the user commands during the navigation process. The results show that the remote user was easily able to move forward and control the wheelchair straight ahead.

Secondly, the experiments were conducted in our laboratory's room. The test area is about 30 square meters. Objects have been placed in the test room to recreate a real home environment. The driving path is more challenging than that in the previous experiment. A path in a figure-eight curve was drawn on the floor using a bright white dashed line, as shown in Figure 15a,b. It was marked to

help the remote user to perform the task and to avoid obstacles in the room. The visual results of this experiment are illustrated in Figure 16. In this experiment, the remote user was asked to drive the wheelchair following the route, which was drawn on the floor. During experiments, all trails in the testing were also video recorded, and the paths were traced and then used for evaluations.

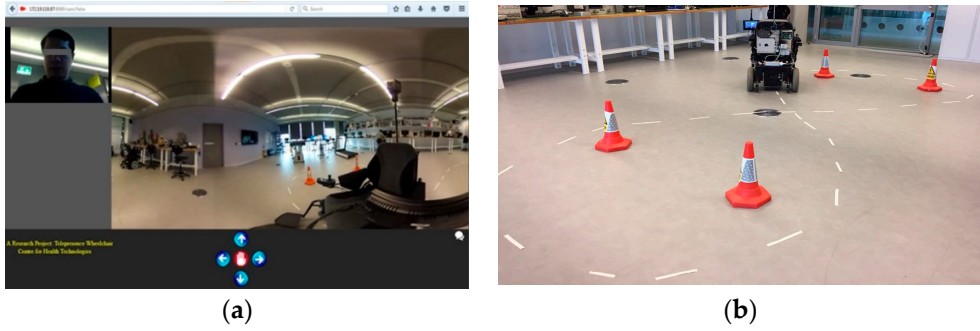

(**a**)　　　　　　　　　　　　　　　　　(**b**)

**Figure 15.** The full field of view obtained at the remote user (**a**) and test path setup with the figure-eight curve (**b**).

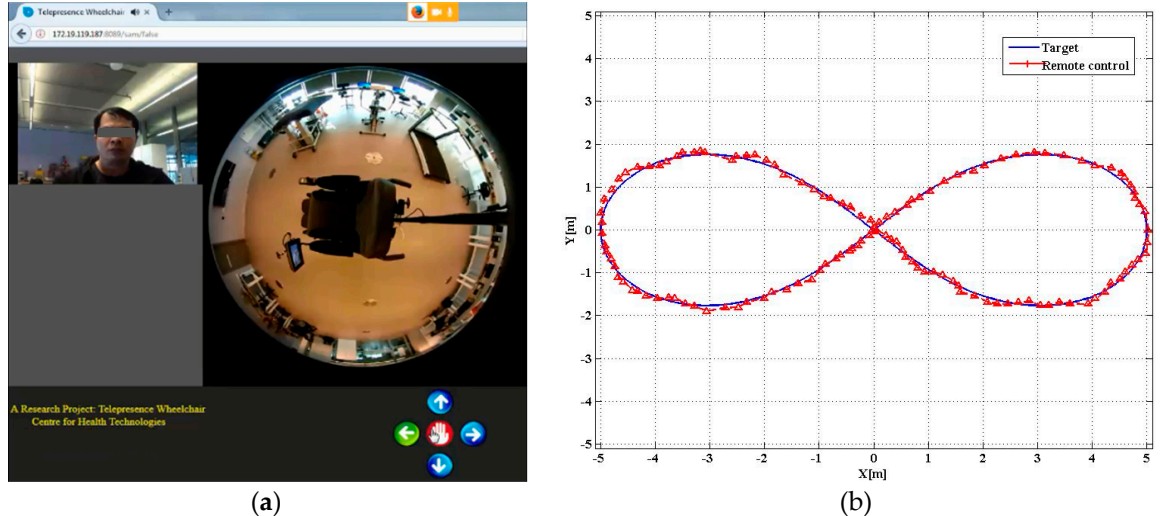

(**a**)　　　　　　　　　　　　　　　　　(**b**)

**Figure 16.** The full top-down view obtained at the remote user (**a**) and the trajectory of the remote control (**b**).

The final experiment presents the remote control performance of the telepresence wheelchair in an outdoor environment, as shown in Figure 17. The test was conducted to validate the effectiveness of the remote control in an outdoor environment.

Experimental results of the remote controlling task in an indoor environment are shown in Figure 16 and an outdoor environment are illustrated in Figure 17. It can be seen from Figures 16 and 17 that the performance of the telepresence wheelchair is very similar to the planning path. The trajectories made by two experiments are very similar to the planning paths with small errors. The results indicate that the wheelchair responds accurately to the control signal, and the performance of the remote control is satisfactory under the design conditions with wireless coverage. These experimental results were found to be acceptable with the desired results.

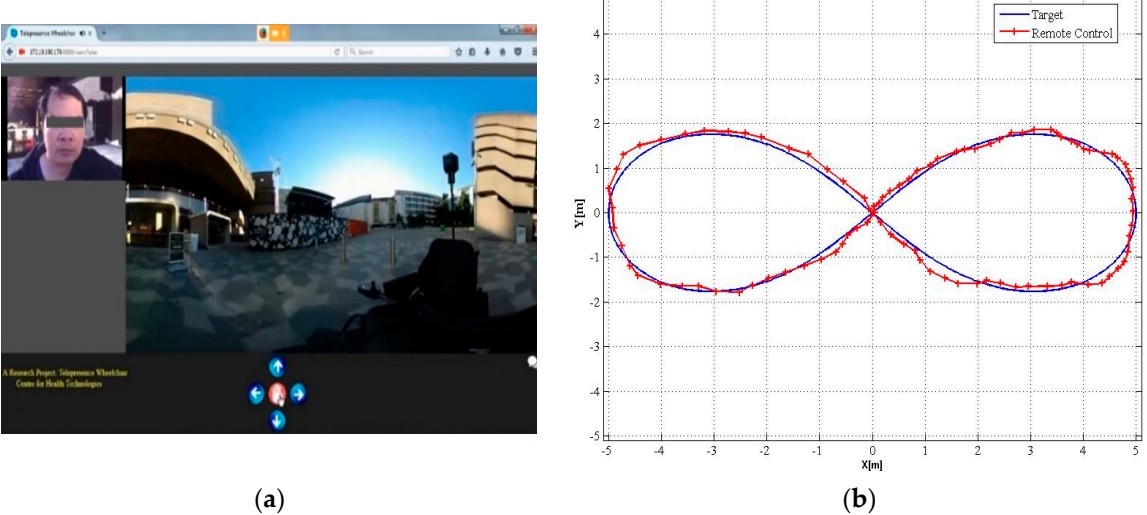

(**a**)             (**b**)

**Figure 17.** The telepresence wheelchair experiments in an outdoor environment (**a**) and the trajectory of the remote control (**b**).

## 4. Discussion

The purpose of this study was to develop a telepresence wheelchair which can provide a 360-degree field of vision using WebRTC. The complete hardware and software implementations of the telepresence wheelchair system, along with the emerging technology in the field of informatics and computer vision have been presented. The experiments were carried out in real-world environments to evaluate system performance. The experimental results are positive compared to the standard recommendation and other recent approaches based on traditional platforms [23]. The results reveal the promise and potential of WebRTC for such a real-time telepresence wheelchair system.

From an implementation point of view, real-time video streaming implementation over wireless networks faces various problems due to both the nature of wireless radio channels and the stringent delivery requirements of media traffic. One of the biggest issues are the unpredictable behaviors of wireless channels due to the fluctuation of the wireless signals and wireless channel conditions. Therefore, one of the primary objectives of video streaming over the wireless network is to deal with the fluctuation of an unknown and dynamic wireless signal. It is important to match the video bitrate to that which the channel can support to improve the quality of video transmission in a wireless network. Thus, it is highly desired that the video streaming scheme can adapt the bit rate according to network conditions.

This study has considered WebRTC that are implemented at the sender and the receiver. The initial results of the first experiment (telepresence wheelchair based on WebRTC) show that the connection time of WebRTC is faster with less latency than the Skype platform. These results are entirely supported by previous studies [20,21,24] which demonstrated real-time telepresence. This result revealed that WebRTC processing is done at the user level, which reduces the delay in processing and connection time. Moreover, the transmission rate of WebRTC is able to adapt to the available bandwidth of the network, whereas Skype platform adjusts the video resolution to adapt to the network condition [20,21,24]. Similar findings were observed by another study reported in [25], which indicated that the WebRTC-based video conference system could perform well on the network less than 1 Mbps of bandwidth capacity. These outcomes supported the results of this study. More importantly, WebRTC is open-source and easily accessible. The convergence capabilities of WebRTC technology can provide flexibility for further development and extension.

During the second experiment (QoE estimation of wireless video streaming using neural networks), a QoE estimation model for wireless video streaming over WebRTC for a telepresence wheelchair is evaluated. The QoE estimation model was developed using neural network principles for video sequences streamed over different wireless network conditions. It was also observed that the model developed can be used for online QoE estimation given measurable network factors such as bit rate,

jitter, round-trip-time, and packet loss rate. The estimated QoE is essential for telepresence wheelchair navigation.

In terms of evaluation of remote control telepresence wheelchair with 360-degree vision, the experimental results in this study show that the telepresence wheelchair can perform in various environments. In the case of traveling straight ahead, the remote user has easily piloted the wheelchair to the target. Moreover, the wheelchair was able to be remotely controlled to follow the figure-eight curve in two different environmental conditions consisting of indoor and outdoor environments. Due to the unique telepresence wheelchair equipped with a fisheye camera for healthcare applications, it is hard to make a comparison between our approaches to previous appropriate works. However, regarding visibility support, we found that using such a fisheye camera could bring considerable benefits for a telepresence wheelchair. It is worth noting that the vast majority of existing telepresence robots in the literature, the installed camera is a traditional one. Therefore, a single view is available only at any one time. The received images of the remote environment are displayed on the screen in one direction view [10,26,27]. In such, these systems had a limited field of view. They are not flexible while operating in the reverse direction. Different from the existing telepresence systems, the novel approach presented in this study based on a 360-degree camera and our method provides a full field of view surroundings in all directions. However, a 360-degree camera is technically more complex and might be more expensive, but it offers superior performance.

During the experiments, we also found that the quality of the received video depends on the network bandwidth. This limitation can be overcome by applying high-speed Internet networks combined with mobile broadband data network to secure network connectivity. Nevertheless, our novel approach has proven the effectiveness of WebRTC for a 360-degree vision telepresence wheelchair, and the use of a dual-fisheye camera along with WebRTC can deliver everything surrounding the wheelchair in 360 degrees. The state-of-the-art technologies would enable various interesting applications, including healthcare support and remote monitoring.

Ultimately, considering the approach of this work, it is clear that the development of the telepresence wheelchair-using WebRTC is a promising concept in the field of assistive technology. The evaluation of the system performance and the findings contribute to the wireless streaming, telepresence system, and real-time application development. Experiments have shown the feasibility of the prototype system. The main contributions of this paper are to provide a concept for the design and implementation of the telepresence wheelchair system for people with disabilities. There are many potential usages of the system to integrate with other healthcare monitoring systems. For future works, the focus work could be on optimization of the hardware and software such that the system can be embedded to accomplish a commercial telepresence wheelchair system.

## 5. Conclusions

This paper has presented the development of a telepresence wheelchair with 360-degree vision. By exploiting the advancements in the field of emerging information technology and digital cameras, efficient technology integration for developing the telepresence wheelchair with 360-degree vision has been explored. Several experiments were conducted to evaluate the developed telepresence wheelchair system. The outcomes of the experimental results were positive. The practical experiments indicated that the proposed technique is successfully streaming and navigating the wheelchair over wireless communication with excellent performance qualities. The overall performance results convince us that the proposed approach of a 360-degree view of the telepresence wheelchair is feasible. Furthermore, a QoE estimation model for wireless video streaming over WebRTC for a telepresence wheelchair was presented. The model was developed using neural network principles for multiple video sequences streamed and bit rates in different network conditions. The developed model can be used for online QoE estimation given measurable network factors which are necessary for the remote user to control the wheelchair towards reliable wireless signal coverage. Overall, the results confirmed the effectiveness of our approach for the telepresence wheelchair with a full field of view. For future works, the focus work could be on optimization of the hardware and software such that the system can be embedded to accomplish a commercial telepresence wheelchair system.

**Author Contributions:** V.K.L.H. investigated the methodologies, developed, implemented, conducted the experiments, analyze the experimental data, and drafted the manuscript. R.C. contributed to the technical, scientific aspects and revision of the manuscript. H.T.N. supervised the study and contributed to the overall research and revision of the manuscript. All the authors contributed to the proofreading and approved the final version of the manuscript.

**Funding:** This work was supported by Australian Research Council under the Discovery Grant DP0666942.

**Conflicts of Interest:** The authors declare no conflict of interest.

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
