# Peer review of "A Telepresence Wheelchair with 360-Degree Vision Using WebRTC"

_applsci, doi:10.3390/app10010369_

Round 1

Reviewer 1 Report

The authors mainly present a telepresence system i.e., transmission of audio and video through internet. They use dual fisheye camera for 360 degrees view. This whole setup is integrated into a wheelchair and everything is working in real time. The paper is well written, clear and presented in a good way. I have few comments,

1. The paper lacks a strong motivation.

Generally, wheel chairs are operated by users who sit on them. So, how far these telepresence wheel chairs are useful and important.

2. Line 75-82
The reason explained in this scenario could be over come by telepresence robots. Using wheel chairs for social interactions are very odd and would effect the interaction in itself. Many robots are built especially for this scenario such as Pepper robot. Researcher MICHAUD François published similar work with telepresence robots which could be found at this site https://introlab.3it.usherbrooke.ca/mediawiki-introlab/index.php/Telerobot and other researcher, Annica Kristofersson worked with elderly people and telepresence robots such as described in this paper: Assessment of Interaction Quality in Mobile Robotic Telepresence: An Elderly Perspective and there are more researchers who are working towards using social robots for social interaction.

3. Line 85-86
Telepresence wheelchair systems are designed not only to provide assistance to the wheelchair users but also to act as a bridge between the participants.

Who are participants in this scenario?

4. Discussion is just a summary of the results and there is no actual discussion. I would like to see some discussion about current robots which use telepresence system and also people who are using autonomous robotic wheelchairs. I would also like to hear about the limitation of the camera which the authors are using in this paper. Similar cameras are previously used in robots such as Giraff telepresence robot. 

Overall the paper is well written, methodology and results are presented in a good fashion. Still, the paper lacks a strong motivation, novelty and discussion. 

Reviewer 2 Report

The paper presents a WebRTC application for remotely controlling a wheelchair equipped with cameras and motor controls. 

The addressed issue is relevant but some issues affect the overall quality of the paper:

As a first general comment, it seems to me that considering a wheelchair as platform is a good choice for addressing healthcare scenarios. But the relevance of this choice is lowered (or even not considered) during the evaluation. The paper should emphasize the characteristics of wheelchair and analysing the aspects related to this specific platform. A deeper analysis of the state of the art on telepresence control is required. This is even more important given that the analysis is on a wheelchair but basically implementing just telepresence platform functions.
Anyway, authors must further investigate existing studies about telepresence solutions (in particular on telepresence solutions evaluation). For instance, authors can consider the work on Giraff platform related to Excite and GiraffPlus EU funded projects: Cesta et al. Long-Term Evaluation of a Telepresence Robot for the Elderly: Methodology and Ecological Case Study. Int. Journal of Social Robotics. 2016. Orlandini et al. ExCITE Project: A Review of Forty-Two Months of Robotic Telepresence Technology Evolution. MIT Presence. 2016 Cesta et al. User needs and preferences on AAL systems that support older adults and their carers. Journal of Ambient Intelligence and Smart Environments. 2018 The WebRTC application is briefly presented. I would encourage authors to provide more details about the architecture and developed applications. As stated by authors, the most crucial issue in deploying telepresence platforms in real environments is WiFi quality. In this regard, the objective of the evaluation is well defined. Nevertheless, the experiments and the evaluation does not seem to "stress" this point. The experiments do not seem to consider "extreme" situations but rather consider controlled situation in which network quality is sufficiently keepe. In this regard, the significance of the evaluation is diminished.
In order to increase the significance of the evaluation, I would strongly suggest to design more complex scenarios in which the evaluation can be performed looking for the "limits" of the solution.
Also, from the text, it is not clear how much is actually stressed the WebRTC application (e.g., stressing the remote control functionalities). 
In this regard, a deeper analysis is highly required. As for QoE training, a question is how much the trained network can be applied in different contexts? This is a quite common problem when using machine learning solution.
In other words, is the learning method flexible enough to adapt different situations? Or should it be adapted (and then repeating training) every time this analysis is performed either under different conditions o different remote users? It seems that MOS analysis is performed in lab. conditions. The most significant results would be required on "into the field" trials. 

In general, the paper seems to require a further effort to provide i) a wider presentation of the WebRTC application, ii) a more thorough analysis of its performance and iii) deeper evaluation and discussion of experimental results.

Round 2

Reviewer 1 Report

Polish little more the motivation and discussion parts

Minor Spell checks:

Line 43, contronlling Line 431, 3560-degree

Reviewer 2 Report

The paper has been improved. Nevertheless, some of the issues raised in my previous review are still there.

As it is, paper seems to require further work for increasing its overall quality.
